# Characterization and Effect of Refining on the Oil Extracted from Durum Wheat By-Products

**DOI:** 10.3390/foods11050683

**Published:** 2022-02-25

**Authors:** Giacomo Squeo, Roccangelo Silletti, Giulia Napoletano, Marcello Greco Miani, Graziana Difonzo, Antonella Pasqualone, Francesco Caponio

**Affiliations:** 1Department of Soil, Plant and Food Science (DISSPA), University of Bari Aldo Moro, Via Amendola, 165/a, I-70126 Bari, Italy; giacomo.squeo@uniba.it (G.S.); roccangelo.silletti@uniba.it (R.S.); graziana.difonzo@uniba.it (G.D.); antonella.pasqualone@uniba.it (A.P.); 2Laboratory R&D Casillo Group, Casillo Next Gen Food srl., Via A. Sant’elia, I-70033 Corato, Italy; giulia.napoletano@casillogroup.it (G.N.); marcello.miani@casillogroup.it (M.G.M.)

**Keywords:** durum wheat oil, bran, milling, edible oil, refining process, tocotrienols, tocopherols, phytochemicals, phytosterols, polar compounds

## Abstract

Durum wheat is one of the most important cereal grains worldwide, used mostly for pasta making and bakery products. The by-products derived for the milling process, although very abundant, have only limited use. The aim of this work was to characterize the oils extracted from the by-products of debranning (DP) and milling processes (MP) of durum wheat and to follow the changes due to the refining process on the minor components. The results showed that DP had significantly higher oil content than MP, but it was characterized by a significantly lower amount of tocols. Polyunsaturated fatty acids content was similar (around 62% of total fatty acids). Consequently, a mixture of DP/MP (60/40 *w*/*w*) was chosen as a basis for further studies concerning the refining process. During refining, carotenoids almost disappeared while tocols were reduced by 24% on average. Free fatty acids, peroxide value, and oxidized triacylglycerols were significantly reduced by refining, while triacylglycerol oligopolymers were significantly higher than the crude oil. Durum wheat oil had an outstanding content of phytosterols and policosanols. Overall, the edible oil obtained from durum wheat after refining could be considered a good source of phytochemicals and could represent a valuable strategy to valorize the by-products from durum wheat mills.

## 1. Introduction

Durum wheat (*Triticum turgidum* L. var *durum* Desf.) is the fundamental ingredient of pasta, bulgur, couscous, and many types of bread typically consumed in the Mediterranean basin and beyond. The total lipid content of durum wheat kernels accounts for about 2.9–3.5% on a dry basis (d.b.) [1], with a fatty acid profile characterized by the prevalence of linoleic acid (about 53%) [2,3]. The latter is an essential fatty acid, which is the precursor of prostaglandins and membrane phospholipids involved in the regulation of blood lipid and cholesterol levels [4,5], not to mention the relevant content of tocopherols, which is another well known and appreciated characteristic of the wheat lipid fraction [6]. About 65% of the wheat lipids are contained in the germ or embryo, and about 15% in the bran, particularly in the aleuronic layer [7]. The rest of the lipid content is distributed in the endosperm (about 20%) [7]. Roughly 2–3% of the grain weight is represented by the germ, and 25% by the bran. Germ and bran represent the main by-products of the milling industry, mostly destined for animal feeding [8].

While the industrial use of germ for oil extraction is well established in the bread wheat chain (*Triticum aestivum* L.), and it is consequently relatively easy to find wheat germ oil at the specialized shops for herbal and health products, durum wheat oil is not marketed yet. The lipids of durum wheat by-products remain therefore largely underutilized, and great potential of improvement still exists regarding their exploitation, also in view of fulfilling the needs of the circular economy.

A recent research interest toward the exploitation of this valuable material is demonstrated by a few scientific articles aimed at characterizing the oil of durum wheat by-products [2,3,9,10]. In these researches, the lipid fraction was generally solvent-extracted by the Soxhlet method, and then the crude oil, which, however, is unsuitable for direct consumption without refining, was characterized. Cardenia et al. [9] analyzed, among other compounds, the sterol composition and observed a prevalence of beta-sitosterol, campestanol, campesterol, and sitosterol. Zarroug et al. [2] and Güven and Kara [3] determined the acidic composition and beta-carotene content in comparison with bread wheat germ oil and found higher levels of beta-carotene in durum than in bread wheat oil. Carotenoid compounds determine the typical yellow color of semolina and, in turn, of pasta and other durum wheat products and are more concentrated in durum than in bread wheat as a result of the genetic selection for high yellow pigment concentration in durum wheat breeding programs [11]. Instead of the Soxhlet extraction, Durante et al. [10] considered the supercritical carbon dioxide (SC-CO_2_) as extraction technology, which is very interesting being eco-friendly but, at the same time, is still too expensive to have a large application. The authors highlighted the richness of durum wheat by-products in vitamin E and carotenoids [10]. It is well known, however, that the refining process may decrease the content of minor compounds present in the starting raw material [12,13]. It would be important, therefore, to assess the extent of this decrease in durum wheat oil.

Since the production of durum wheat oil could represent an added value for the durum wheat milling industry, the aim of the present research was two-fold: (i) characterize the oils obtained from two different durum wheat by-products (from debranning and from milling processes) and (ii) evaluate the changes that occur during the oil refining process.

## 2. Materials and Methods

### 2.1. Reagents

Potassium iodide, glacial acetic acid, chloroform, sodium thiosulfate, sodium hydroxide, starch, β-carotene, α-tocotrienol, α-tocopherol, and α-cholestanol were purchased from Sigma Aldrich (Milan, Italy). Petroleum ether, diethyl ether, ethanol, tetrahydrofuran, *n*-hexane, citric acid, and sodium sulfate anhydrous were purchased from Carlo Erba (Rodano, Italy). Bleaching earth was purchased from Clariant (Milan, Italy).

### 2.2. Sampling

All samples were supplied by Molino Casillo spa (Corato, Italy) and Figure 1 shows the flow chart of the experimental trials.

In particular, two different durum wheat bran by-products, one from the debranning process (DP) and the other from the milling process (MP), were considered. In particular, DP was composed of the second and third debranning fractions; three was the total number of fractions consecutively detached from the wheat kernel, where the first was the outmost, discarded, and the third was the innermost. MP consisted of milling middlings with 7.1 g/100 g fiber and 3.8 g/100 g ash. Five samples (*n* = 5) per by-product type were furnished and subjected to oil extraction and subsequent analysis, as reported in the next sections.

Moreover, two durum wheat oils (*n* = 2), sampled from large batches and representative of those extracted from a selected mixture of by-products (60/40 *w*/*w*, DP/MP), were subjected to the refining process. Two samples per each of the refining steps (crude, degumming, neutralization, bleaching, deodorizing) were supplied by Casillo Next Gen Food srl. (Corato, Italy), packaged in a glass bottle hermetically closed and kept at −18 °C until they were analyzed.

### 2.3. Oil Extraction

Each durum wheat bran by-product was subjected to lipid extraction in the laboratory by the Soxhlet method using diethyl ether (34–35 °C) as a solvent for 6 h.

Crude durum wheat oils were obtained by a solvent extracted with *n*-hexane (1:5, *v*/*v*; Carlo Erba, Rodano, Italy) using an ultrasonic extractor (STEEL-1000-DG, Unitech srl., Vigonza, Italy) at room temperature for 6 h, starting from a mixture of the DP and MP dried in a thermostatic oven (Universalschrank UN 260, Memmert, Schwabach, Germany) for 90 min at 105 °C and 60% ventilation until to reach a relative humidity of 6–8% determined by thermobalance (MA 50/1. R Moisture Analyzer). At the end of the extraction cycle, the oil–hexane mixture was processed in a vacuum evaporator (IST15, I.S.T. Italia Sistemi Tecnologici SPA, Modena, Italy) at 90 °C and 0.20 bar for 3 h to obtain crude durum wheat oil. Two different crude durum wheat oils, each representative of a large amount of durum wheat by-product, were subjected to each refining step

### 2.4. Refining Process

Each crude durum wheat oil was subjected to refining in duplicate using a laboratory-scale plant. The refining system consisted of 4 L glass vessels equipped with a heating system, variable-speed stirring, vacuum and nitrogen purge systems, and steam deodorization apparatus.

#### 2.4.1. Degumming

Crude durum wheat oil at 60 °C was added with 5% of water, acidified with citric acid (1%, *v*/*v*), and the mixture was stirred for 15 min to gum formation and aggregation. Then, the mixture was centrifuged at 4500 rpm for 15 min (SL 8/6R, ThermoFisher Scientific, Milan, Italy) to remove the hydrated phospholipids and other water-soluble impurities.

#### 2.4.2. Neutralization

Degummed oil was neutralized to remove the free fatty acids. In particular, oil at 60 °C under vacuum was added to alkali and calculated based on the stoichiometric amount plus 10% excess. Then, the mixture was heated until 80 °C and neutralized under continuous stirring for 10 min and subsequently centrifuged at 4500 rpm for 15 min (SL 8/6R, ThermoFisher Scientific, Milan, Italy) and washed with hot water (75 °C) until the washing water is neutral to remove the soaps completely.

#### 2.4.3. Bleaching

Neutralized oil was treated with 3% (*w*/*w*) bleaching earth (Tonsil^®^ Optimum 210 FF, Clariant, Milan, Italy) and under stirring bleached at the following conditions: vacuum 50 Mbar, temperature 80 °C, time 30 min. Then, the bleaching earth was removed by filtering the oil under a vacuum after cooling to 60 °C.

#### 2.4.4. Deodorization

Bleached oil at a residual pressure value < 1 mm Hg was heated under nitrogen flow until 210 °C and then subjected to deodorization at a constant steam flow rate of 10 mL water/hour for 120 min.

### 2.5. Analytical Determinations

#### 2.5.1. Routine Analyses and Fatty Acids Composition

The determinations of free fatty acids (FFA) and peroxide value (PV) were carried out as prescribed by the EEC Regulation no. 2568/91 [14] and subsequent amendments and integrations. The determination of fatty acid composition was carried out after sample transesterification with KOH 2N in methanol [14] by a GC (Agilent 7890A gas chromatograph, Agilent Technologies, Santa Clara, CA, USA) equipped with an FID detector (set at 220 °C) and an SP2340 capillary column of 60 m × 0.25 mm (i.d.) × 0.2 mm film thickness (Supelco Park, Bellefonte, PA, USA). The identification of each fatty acid was carried out by comparing the retention time with that of the corresponding standard methyl ester (Sigma-Aldrich, St. Louis, MO, USA). The amount of single fatty acids were expressed as area % with respect to the total area.

#### 2.5.2. Carotenoids Determination

Carotenoids are spectrophotometrically measured (Cary 60 Agilent spectrophotometer, Milan, Italy) at 449 nm after dissolving 0.25 g of oil in hexane to the final volume of 10 mL [15]. The results are expressed as mg β-carotene per kg of oil by using an external calibration curve of β-carotene.

#### 2.5.3. Tocotrienols and Tocopherols Determination

Tocotrienols and tocopherols are determined by RP-UHPLC-FLD (Dionex Ultimate 3000 RSLC, Waltham, MA, USA) [16]. The sample was weighted (0.04–0.12 g), dissolved in 1 mL of 2-propanol, and filtered. Then 20 μL were injected into the UHPLC system equipped with an HPG-3200 RS Pump, a WPS-3000 autosampler, a TCC-3000 column compartment, and an FLD-3400RS fluorescence detector using a Dionex Acclaim 120 C18 analytical column (150 mm × 3 mm i.d.) with a particle size of 3 μm (Thermo Scientific, Waltham, MA, USA). The mobile phase consisted of a mixture of acetonitrile and methanol (1:1 *v*/*v*) at a constant flow rate of 1 mL/min in isocratic elution. The FLD was set at an excitation wavelength of 295 nm and an emission at 325 nm. The quantification was reached by means of the external calibration method based on calibration curves obtained for α-tocotrienols and α-tocopherols. The results were expressed as mg of tocotrienols and tocopherol per kg of oil.

#### 2.5.4. Polar compounds Separation and High-Performance Size-Exclusion Chromatography (HPSEC) Determination

The polar compound (PC) analysis was carried out by following the method described in Caponio et al. [17] with the same modifications. Briefly, oil was weighted (0.5 g), dissolved in the eluent (mixture petroleum ether/diethyl ether; 87/13, *v*/*v*), and firstly, the nonpolar fraction was recovered. The amount of total polar compounds was calculated as the difference between the weight of the sample added to the column and that of the nonpolar fraction eluted and expressed in g per 100 g. Subsequently, the polar compounds were extracted by performing a second elution with diethyl ether. The ether was then removed, and the polar fraction was recovered in tetrahydrofuran (THF) and subjected to HPSEC analysis by using THF as eluent at 1 mL/min flow rate. The HPSEC system consisted of a series 200 pump (Perkin-Elmer, Norwalk, CT, USA) with Rheodyne injector, a 50 μL loop, a PL-gel guard column (Perkin-Elmer, Beaconsfield, UK) of 5 cm length and 7.5 mm i.d., and a series of two PL-gel columns (Perkin-Elmer, UK) of 30 cm length and 7.5 mm i.d. each. The columns were packed with a highly cross-linked styrene-divinylbenzene copolymer with particles of 5 μm and a pore diameter of 500 Å. The detector was a series 200 refractive index (Perkin-Elmer, USA). Polar compounds were identified by polystyrene standards (Supelco, Milan, Italy) as reported in a previous paper. For quantitative determination of the single polar compounds, known amounts of triacylglycerol oligopolymers (TAGP), oxidized triacylglycerols (ox-TAG), and diacylglycerols (DAG) were obtained by preparative gel permeation chromatography of PC derived from refined peanut oil and then used as standards in the HPSEC. The results were reported as g of each fraction per 100 g of oil.

#### 2.5.5. Phytosterols and Policosanols Determination

The sterol composition was determined as described in Miazzi et al. [18]. Briefly, about 5 g of oil was added with α-cholestanol as internal standard and subjected to saponification with a solution of KOH in ethanol (2 N) under heating. The sample was transferred in a separating funnel and washed three times with ethyl ether to collect the unsaponifiable fraction. The etheric phase was neutralized and filtered by sodium sulfate anhydrous and dried. The sterol fraction, resuspended in chloroform (5%), was separated from the unsaponifiable matter by tin layer chromatography and then recovered, filtered, and silanized. Finally, about 1 mL of the solution was injected into the GC system (Agilent 7890A) using a capillary column HP-5 30 m × 0.32 mm (i.d.) × 0.25 mm film thickness (Agilent Technologies, Santa Clara, CA, USA). The injector temperature was 290 °C with a split ratio of 1:25. The identification was carried out by comparing the retention time with those reported in the official method [14]. Single sterols content was reported as area percentage with respect to the total sterol area, while the total content was calculated using the internal standard method and expressed as mg/kg. The policosanol determination was carried out by gas-chromatographic analysis as reported in Harrabi et al. [19].

### 2.6. Statistical Analysis

The results were reported as mean ± standard deviation. One-way analysis of variance (ANOVA) followed by Bonferroni post hoc test for multiple comparisons was used to highlight significant differences at α = 0.05. Statistical analyses were carried out by Minitab 17 (Minitab Inc., State College, PA, USA).

## 3. Results and Discussion

### 3.1. Characterization of the Oils from Debranning and Milling Fraction

The total oil content and its characteristics are key aspects to be monitored to evaluate the possible and feasible exploitation of raw materials for oil extraction. DP and MP fractions showed an oil content equal to 6.0% (±0.3%) and 4.8% (±0.4%), respectively. Significantly higher oil content was found in DP, showing on average about 1.2 g of oil more than MP per 100 g of raw material. This result reflected the higher share of aleurone layer and germ particles, richer of lipids than endosperm [7], in the DP than in MP. The obtained data were also in accordance with the findings of Cardenia et al. [9], both for total oil content and for distribution of oil among the two different fractions considered.

The fatty acid composition of the extracted oils is reported in Table 1. Durum wheat oils were rich in polyunsaturated fatty acids (PUFA) (61.71% and 62.46%, respectively, for DP and MP), which represent the most abundant class of fatty acids, with linoleic acid (C18:2) being the most representative (on average around 91% of total PUFA) followed by the linolenic acid (C18:3), and no significant differences were highlighted between the fractions. Considering saturated fatty acids (SFA), a significantly higher content was observed for MP oil which, on the other hand, showed significantly lower values of monounsaturated fatty acids (MUFA). The most representative fatty acids were palmitic acid (C16:0, roughly 90% of total SFA) and oleic acid (C18:1, 97% of total MUFA), respectively. Our data were in good accordance with those reported by Zarroug et al. [2] and Cardenia et al. [9], even though they found slightly lower values of PUFA and higher SFA.

Table 2 reports the content of total carotenoids, tocotrienols, tocopherols, and their sums in oils obtained from debranning (DP) and milling processes (MP). Regarding carotenoids, no significant differences were highlighted between the examined fractions, although, on average, DP showed a higher amount than MP. Kumar and Krishna [20] found a lower amount of total carotenoids in wheat bran oil and a higher amount in wheat germ oil. Regarding the content of the total tocols, it was significantly different between the oils. DP was richer in total tocotrienols (about 23% higher), while, in contrast, it had a significant minor content of tocopherols with respect to MP (about 56% lower). Therefore, the total amount of vitamin E was significantly higher in MP with respect to DP. The α and β were the most abundant isomers among the others (data not shown), with the latter being the predominant tocotrienol (about 85%) and the former the predominant tocopherol (about 60%). To the best of our knowledge, no data were present in the literature regarding the content of the tocols found in durum wheat oils. However, considering other oils, a similar amount was detected in wheat bran oil [20], while in wheat germ oil, tocopherol content can reach up to 2500 mg/kg and α-tocopherol (60%) is predominant [21].

According to the reported results, durum wheat oil could be considered a good source both of essential fatty acids and bioactives, with both being fundamental for human health [22,23,24,25,26,27] and in the oxidation of fats and oils acting as free radical scavenging [28]. Following the reported outcomes and considering that MP was richer in total tocols while DP was richer in oil, a mixture of these by-products (60/40 *w*/*w*, DP/MP) was subjected to the oil extraction process for further studies. This oil, extracted in duplicate, was then subjected to the refining process.

### 3.2. Effect of the Refining Process on Durum Wheat Oil Quality

Most vegetable oils are not suitable for direct consumption and must undergo a refining process. In fact, depending on the raw material and the oil extraction technology applied, crude oils might contain, among the others, free fatty acids, polar lipids, pigments, volatile compounds, contaminants, or autoxidation products. The refining process aims at removing such undesirable compounds by subsequent steps that can be summarized in (i) degumming, (ii) deacidification or neutralization, (iii) bleaching, (iv) deodorization. More details about the refining process of vegetable oils could be found elsewhere [29].

Refining affects the physico-chemical and quality parameters of the product, and Table 3 reports the results obtained during durum wheat oil refining. The amount of FFA in crude oil (11.58%) significantly decreased during refining, being practically eliminated at the end of the refining process. PV was also influenced by refining, showing a decrease after degumming and reaching the lowest absolute value at the end of the refining process. TAGP significantly rose after the bleaching and deodorization steps. On the opposite, ox-TAG were lower in bleached and refined oil, reaching the significantly lowest values in the latter. The amount of DAG did not change significantly during the process, as expected. Overall, the total amount of PC decreased significantly from the crude oil to the refined one reaching, in this case, a final value of around 7 g/100 g oil. Overall, the degree of hydrolytic degradation is similar to those detected in olive-pomace oils [30,31], while the extent of oxidative degradation is comparable to those detected in refined olive oil and definitely lower to the levels found in olive-pomace oils and seed oils [30,32,33]. It is worth noting that the products of triacylglycerol polymerization, at high amounts, could have harmful effects on the consumer’s health [34,35], as well as a pro-oxidative activity [36,37,38].

The levels of FFA and PV of the refined durum wheat oil were similar to those observed in previous studies regarding refined industrial oils of other botanical species [30,32,39,40]. The FFA content of the crude oil from bread wheat germ is usually high (5–25% is typical) and varies depending on the conditions of germ storage and oil extraction [41]. For what concerns the evolution of the oil’s characteristics during refining, the trends observed for FFA and PV are in accordance with those reported by other authors [42]. Obviously, the neutralization step drastically reduced the level of FFA, as expected. Concerning PV, a sort of bell-shaped trend was observed in our study and confirmed in others [43], showing a maximum PV at the neutralization step. According to the Codex Alimentarius [44], the limit for FFA and PV in refined oils is 0.6 mg KOH/g oil and 10 meq O_2_ per kg, respectively. In our experiments, the final value of PV was quite lower than the limit, while the FFA value was slightly higher and equal to 0.8 mg KOH/g oil. However, it should be considered that the process was carried out at a lab scale, and the results could be linked to the minor performance of the used system. Nonetheless, the value is very close to the maximum allowed, so it could be assumed that more performant refining plants could produce oils that could easily match the desired values.

Polar compounds are principally constituted by triacylglycerol oligopolymers, oxidized triacylglycerols, and diacylglycerols. The quantification of these compounds by HPSEC analysis was already successfully applied to (i) determine the real level of oxidative and hydrolytic degradation of oils, either refined or subjected to treatments requiring high process temperatures [31,39,45]; monitor the changes occurring in oil during frying [46,47,48,49]; assess the quality of the lipid fraction of various foods [50,51,52,53]; and discriminate among extra virgin olive oils, refined, and mild deodorized ones [54]. TAGP are considered a reliable index of secondary oxidative degradation of oils because of their development during the bleaching and especially the deodorization step in the refining process [31,33,55] and are stable and not influenced by processing conditions. Moreover, ox-TAG could provide information about the primary oxidation level of oil and comprise all forms of triacylglycerols oxidation and, finally, DAG constitutes a reliable parameter to assess the real level of hydrolytic degradation of refined oils.

While on the one hand, refining removes unwanted negative products from oils and fats, on the other hand, it could cause a depletion in the minor compounds. With this regard, the evolution of tocols during the process was followed, and the results are reported in Figure 2. First, it could be observed that in the crude blend oil, there was a predominance of tocotrienols that accounted for about 74% of the total vitamin E content. During the refining process, the level of tocotrienols remains similar without any significant difference between the sample till the bleaching step, from which a significant decrease was observed. The final deodorization did not cause other significant changes. In the final refined oil, a total decrease of around 19% was observed with respect to the crude oil. A similar trend was observed for the total tocopherols. In fact, till the neutralization step, similar contents with no significant differences were observed between the samples. Thereafter, bleaching caused a significant reduction in tocopherols, whose values remained almost constant after the last deodorization step. The total decrease with respect to the original oil was about 36%. Obviously, the total vitamin E content followed the same trend experiencing a significant reduction of 24% due to the whole refining process. Overall, the evolution of total vitamin E was best described by a second-order polynomial (R^2^_adj_ = 0.77), confirming the parabolic trend during refining (Appendix A). Wang and Johnson [41], during the refining of wheat germ oil, highlighted that degumming, neutralization, and bleaching did not significantly influence tocopherol contents, while deodorization conditions had a great impact. Carotenoids (data not shown) were almost totally reduced after refining as a consequence of the bleaching step [12].

Together with antioxidants, phytosterols and policosanols are others important classes of bioactives involved in the balance of cholesterol concentrations by reducing intestinal absorption [23,56]. Cereal by-products, in addition to plant-origin foods and vegetable oils, contain significant amounts of phytosterols and could represent a potential source of these health-enhancing compounds. It is well known that bran fractions and germ oil are some of the best sources of sterols [57]. Policosanols, instead, are a mixture of long-chain aliphatic alcohols (C22–C34) obtained from different natural food, such as beeswax, sugarcane, rice bran, and wheat germ [58].

Table 4 reports the mean content of phytosterols and policosanols detected in the refined durum wheat oil obtained from debranning and milling process. The high content of phytosterols was found in refined durum wheat oil, comparable to those detected by other authors in rice bran and wheat germ but decisively higher than the content found in oat bran and oat hull [59] or in more common seed oils [60]. Among sterols, sitosterol and campesterol, and their saturated counterparts sitostanol and campestanol, accounted for about 80% of the total sterols, similarly to what was found by Jiang and Wang [59]. Regarding policosanols, the amount detected in refined durum wheat oil was comparable to those found in insect wax but decidedly higher than the content found in commercial vegetable oils [58].

## 4. Conclusions

Durum wheat mills produce large amounts of by-products that could be valorized, as it is rich in valuable phytochemicals. The results obtained in this study show that the debranning and the milling fractions have an exploitable amount of residual oil, which is very rich in tocopherols, tocotrienols, phytosterols, and polyunsaturated fatty acids. During refining, tocols were significantly but not dramatically reduced with respect to the crude oil. In the refined oil high amount of phytosterols and policosanols were found while free fatty acids and peroxides were removed. These results gave useful insights for the subsequent optimization of the industrial oil extraction and refining processes from durum wheat by-products.

## Figures and Tables

**Figure 1 foods-11-00683-f001:**
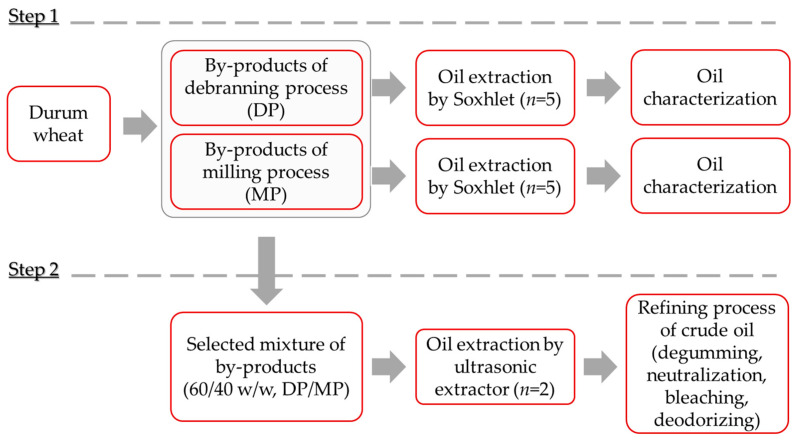
Flow chart of the experimental trials.

**Figure 2 foods-11-00683-f002:**
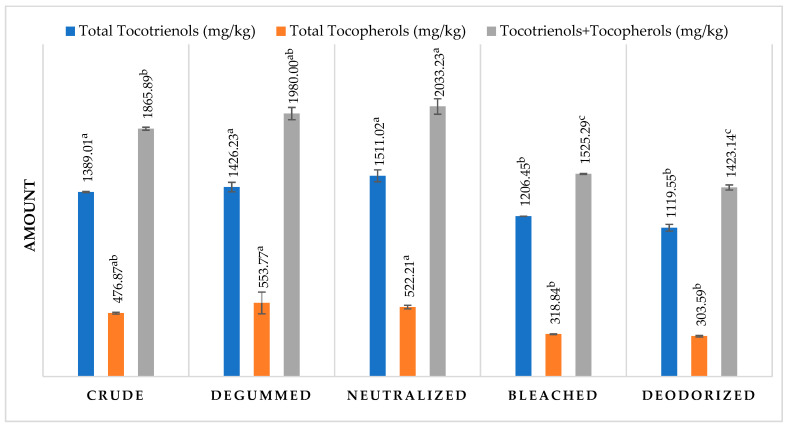
Tocotrienols and tocopherols content of the extracted crude durum wheat oil during refining. Mean values (*n* = 2), standard deviations, and results of statistical analysis. Different letters for each parameter indicate significant differences (*p* ≤ 0.05).

**Table 1 foods-11-00683-t001:** Percentage of saturated, monounsaturated, and polyunsaturated fatty acids (SFA, MUFA, PUFA) of the lipid fraction obtained from debranning (DP) and milling processes (MP). Mean values (*n* = 5), standard deviations, and results of statistical analysis.

Fatty Acids (%)	DP	MP
SFA	17.28 ± 0.51 ^b^	18.67 ± 0.37 ^a^
MUFA	21.01 ± 0.52 ^a^	18.86 ± 1.81 ^b^
PUFA	61.71 ± 0.12	62.46 ± 1.51

Different letters in rows indicate significant differences (*p* ≤ 0.05).

**Table 2 foods-11-00683-t002:** Total content of carotenoids, tocotrienols, tocopherols, and their sums of the lipid fraction obtained from debranning (DP) and milling processes (MP). Mean values (*n* = 5), standard deviations, and results of statistical analysis.

Compounds (mg/kg)	DP	MP
Total carotenoids	42.44 ± 1.22 ^a^	37.88 ± 5.07 ^a^
Total tocotrienols	1336.33 ± 66.56 ^a^	1085.76 ± 121.66 ^b^
Total tocopherols	461.77 ± 31.49 ^b^	1050.63 ± 117.54 ^a^
Tocotrienols + tocopherols	1798.10 ± 31.49 ^b^	2136.39 ± 101.28 ^a^

Different letters in rows indicate significant differences (*p* ≤ 0.05).

**Table 3 foods-11-00683-t003:** Hydrolytic and oxidative degradation parameters measured on the durum wheat oil during the refining process. Mean values (*n* = 2), standard deviations, and results of statistical analysis.

Parameters	Crude Oil	Degummed Oil	Neutralized Oil	Bleached Oil	Deodorized Oil
FFA (%)	11.58 ± 0.31 ^a^	10.16 ± 0.38 ^b^	0.64 ± 0.01 ^c^	0.40 ± 0.04 ^c^	0.32 ± 0.04 ^c^
PV (meq O_2_/kg)	4.45 ± 0.65 ^b^	3.89 ± 0.16 ^b^	7.65 ± 0.35 ^a^	4.78 ± 0.07 ^b^	2.05 ± 0.07 ^c^
TAGP (%)	0.14 ± 0.02 ^c^	0.15 ± 0.01 ^c^	0.13 ± 0.01 ^c^	0.26 ± 0.01 ^b^	0.41 ± 0.01 ^a^
ox-TAG (%)	2.28 ± 0.22 ^a^	2.21 ± 0.08 ^a^	2.63 ± 0.12 ^a^	2.15 ± 0.11 ^ab^	1.58 ± 0.06 ^c^
DAG (%)	5.34 ± 0.25 ^a^	5.33 ± 0.21 ^a^	5.29 ± 0.04 ^a^	5.11 ± 0.12 ^a^	5.09 ± 0.09 ^a^
PCs (%)	19.34 ± 0.15 ^a^	17.86 ± 0.68 ^a^	8.69 ± 0.15 ^b^	7.92 ± 0.06 ^b^	7.39 ± 0.13 ^b^

FFA, free fatty acids; PV, peroxide value; TAGP, triacylglycerol oligopolymers; ox-TAG, oxidized triacylglycerols; DAG, diacylglycerols; PCs, polar compounds. Different letters in rows indicate significant differences (*p* ≤ 0.05).

**Table 4 foods-11-00683-t004:** Mean values of the sterolic composition and total policosanols of the refined durum wheat oil.

Compounds	Mean Values
Cholesterol	0.2%
24-Methylenecholesterol	0.7%
Campesterol	13.1%
Campestanol	15.1%
Stigmasterol	1.8%
∆-7-Campesterol	1.4%
Clerosterol	0.5%
β-Sitosterol	34.8%
Sitostanol	16.6%
∆-5-Avenasterol	7.9%
∆-7(9,11)-Stigmastadienol	1.9%
∆-5,24-Stigmastadienol	1.1%
∆-7-Stigmastenol	1.5%
∆-7-Avenasterol	3.4%
Total sterols (mg/kg)	20,975
Total policosanols (mg/kg)	754

## Data Availability

Data is contained within the article.

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
