# Peer review of "Characterization and Effect of Refining on the Oil Extracted from Durum Wheat By-Products"

_foods, 2022, doi:10.3390/foods11050683_

Round 1

Reviewer 1 Report

The research presented in the manuscript foods-1583546 " Chemical characterization of durum wheat oil during refining" has the hallmarks of innovation. It aimed to characterize the oils extracted from the by-products of debranning (DP) and milling processes (MP) of durum wheat and to follow the changes due to the refining process on the minor components. The subject of this study deals well with the scope of the journal and the design of the study is acceptable, however, the quality of the manuscript and the scientific value is quite low. The comments provided below may be helpful in improving this manuscript.

Main comments:

  1. Title - indicate that the by-products were the research material
  2. Both, in the abstract and throughout the text, the abbreviations should be explained where they appear for the first time (not only in the Tables and Figures captions).
  3. Materials and Methods

- add a separate paragraph indicating all standards and reagents suppliers;

- materials analysed were not clearly described; I suggest providing the scheme presenting the sample preparation instead of the long description (paragraph 2.1);

- the by-products of production processes but not the processes it selves were the study material - the research materials should be precisely indicated. The abbreviations DP and MP should be defined properly and then used in the text, tables, and figures. Meanwhile, Figure 1 shows the total lipid content in the debranning and milling process which is inappropriate because in Figure1 the content of lipids in the by-products was indicated

-2.4.1. Routine analyses and fatty acids composition - add details on how was the FA composition analysed? If GC, add all necessary information;

- line 131: the EEC regulation need to be indicated in the references

- Carotenoids determination: add details about the spectrophotometer model;

- Line 134: the cited [Makhlouf et al., 2018] is missing in the references list;

- Line 135: "…as mg β-carotene per kg" – of what?

- Line 182: Commission Regulation (EEC) No. 2568/91 – not indicated in the references list

- The statistical analysis: two replication is rather a small number of repetitions … this is a very weak point of this research

  1. Results - general comments:

The manner of the presentation of the results is unusual, in fact, results description "disappears" in the too excessive discussion, which is sometimes redundant or too obvious;

Table and Figure captions are not informative and neet to be rewritten;

- Besides the poorly characterized composition of by-products (only the total lipids content), it was presented as a separate Figure1 that lacks the description of the Y-axis and the unit; instead the Figure, this information could be simply included in the text

-Tab. 1: how were the data expressed? Unit?

-Fig 2: instead of two not very informative Figures better to present the results in one table (as the Av value and SD); besides, why were the detailed content of tocopherols (α, β, γ, δ) and tocotrienols not analysed in this study if this research aimed to provide a chemical characterization of the oil extracted from durum wheat by-products?

-Lines 255-258: “To the best of our knowledge, no data were present in literature regarding the content the tocols found in durum wheat oils. A similar amount was detected in wheat bran oil [26] while in wheat germ oil tocopherol content can reach up to 2500 mg/kg and α-tocopherol (60%) is predominant” - this is unclear, as the authors compare their results with the data available in the literature

-Line 280: was it a durum wheat germ oil? Why this information appears at this point and not in the description of the research material? According to information in section 2.1  “two durum wheat oils (n = 2), sampled from large batches and representative of those, extracted from a selected mixture of by-products (60/40 w/w, DP/MP)”- clarify these inaccuracies

-Lines 293-295: how the significant increase in PV be explained?

-Tab. 2 and Fig. 3: among the refining processes, deodorization was mentioned in the 2.3 section. Why the results are not presented?

Reviewer 2 Report

The manuscript entitled "Chemical characterization of durum wheat oil during refining" is good, however, here are some comments for improvement:

References should be checked, for example on page 3 line 134, the reference is unnumbered.

In general, the quality of the Figures should be improved, I suggest that authors use programs like Origin, etc. In addition to adding a "y" axis with values.

Statistical analysis should be overwritten in numbers.

Round 2

Reviewer 1 Report

The quality and scientific value of the revised manuscript (foods-1583546) by Giacomo Squeo and co-authors was meaningfully improved. The authors took into account all reviewer comments. The manuscript in its current form is ready for publication.

Author Response

Thank you